# Can welfare states buffer technostress? Income and technostress in the context of various OECD countries

**Ann S. Lauterbach** *⊙, **Tobias Tober**⊙, **Florian Kunze**⊙, **Marius R. Busemeyer**⊙

Department of Politics and Public Administration, University of Konstanz, Konstanz, Baden-Wuerttemberg, Germany

⊙ These authors contributed equally to this work.

* ann-sophie.lauterbach@uni-konstanz.de

## Abstract

Many workers are experiencing the downsides of being exposed to an overload of information and communication technology (ICT), highlighting the need for resources to cope with the resulting technostress. This article offers a novel cross-level perspective on technostress by examining how the context of the welfare state influences the relationship between income and technostress. Showing that individuals with higher income experience less technostress, this study argues that the welfare state represents an additional coping resource, in particular in the form of unemployment benefits. Since unemployment benefits insure income earners in the case of job loss, the negative effect of income on technostress should increase with higher levels of unemployment generosity. In line with these expectations, empirical results based on original survey data collected in collaboration with the OECD show that the impact of income on technostress varies across welfare state contexts. Implications for public health and policymakers are being discussed.

## Introduction

The constant usage of technology characterizes the working environment for employees in the OECD world in many sectors and occupations. In recent representative data from Germany, for example, Information and Communication Technology (ICT) exposure is prevalent among 92% of employees, with work-related ICT use being common for all age groups and occupations [1]. Despite many personal and organisational benefits from an increasingly digitalised workplace, these developments can also create perceptions of limited resources and uncertainties. In the organisational behaviour and information systems literatures this phenomenon has been labeled as technostress, which can be defined as an individual's "struggle to deal with constantly evolving ICTs and the changing cognitive and social requirements related to their use" [2, p. 303]. Thus, workplaces with increasing technology tools might begin to undermine employees' productivity and lead to unwanted technology overload [3]. Stress in using ICT arises when there is a high dependency on ICT, a gap between the workers' knowledge of ICT and what is required, or when there is a change in the work culture due to the use of technology [4].

**Funding:** The authors acknowledge the funding by the Deutsche Forschungsgemeinschaft (DFG—German Research Foundation) under Germany's Excellence Strategy (Grant Number EXC2035/1-390681379). The funders had no role in study design, data collection and analysis, decision to publish, or preparation of the manuscript.

**Competing interests:** The authors have declared that no competing interests exist.

From existing individual-level research, we know that technostress impairs important employee outcomes such as mental health, physical health, and workability [1, 2, 5–7]. Additionally, technostress is negatively related to worker productivity [3, 8–10], organisational commitment [4], and positively to turnover intention [11].

Despite these primarily negative effects of technostress on work outcomes, it is an open question if all employees perceive similar levels of technostress or if socioeconomic backgrounds lead to variations in technostress perceptions. Prior studies have, for instance, assessed the role of education [4, 12, 13], job position [8, 14], and socioeconomic status as measured by skill levels [15, 16].

However, the results concerning the relationship between socioeconomic position and technostress perceptions remain inconclusive. While some argue that perceived technostress might be lower for individuals with higher formal education [4], other research points in the opposite direction [17]. In addition, some studies find no differences in the level of technostress across different socioeconomic groups [1, 12, 18]. Most of these studies use various determinants of socioeconomic status interchangeably. Yet, not explicitly distinguishing socioeconomic factors like education and income limits the interpretation of the empirical results [19–21]. Moreover, the review by Borle *et al.* [20] shows that many studies on technostress use samples that only represent occupations with high socioeconomic status, pointing to likely problems due to sampling bias [20]. Further, the direct link between income and technostress has so far only been examined in studies focusing on very specific techno-invasion facets, e.g., compulsive app [22] or social media usage [23]. However, this perspective so far neglects the multi-faceted construct of technostress also including the dimensions techno-overload, techno-complexity, techno-insecurity, and techno-uncertainty as in the original measure by Tarafdar *et al.* [2]. Therefore, the overall ambition of this study is to extend the literature on the socioeconomic antecedents of multi-faceted perceived technostress by using large-scale, representative data that the authors collected in cooperation with the OECD.

As for our specific contributions, the first is to focus on individual income as an important but understudied variable in the perception of technostress. To our knowledge, none of the extant studies examining the relationship between socioeconomic status and perceived technostress examine the effect of income (for a review, see [20]). Conceptually, income indicates a person's current living condition, e.g. that an individual has a well-paid job and thus financial security. In our study, we theorize that income plays a key role in the relationship between technological development and higher perceptions of technostress. Based on Lazarus's [24] seminal theory of stress and coping, which has recently been extended to the field of technostress [25, 26], we assume that income is a resource that can help workers deal with technological changes and reduce their perceptions of stress.

The second and main contribution is to pay more attention to the role of macro contexts, in particular the welfare state. The psychological literature on the determinants of technostress mentioned above focuses on micro-level antecedents of technostress but does not (yet) acknowledge the potential role of macro-level contexts. In contrast, scholarship in the domain of welfare state research has so far not been concerned with health outcomes related to technology but rather focuses on the general association between health outcomes and welfare state regimes [27–29]. In a similar vein, some research in comparative political economy has started to explore risk perceptions associated with technological change [30–32] as well as technology-related anxiety [33], building on earlier literature focusing on labour market risk and its implication for social policy [34–36], but this body of scholarship does not discuss technostress either. We combine these so far separated literatures and, thus, use the potential for fruitful interactions between organisational psychology and welfare state research.

To summarize and provide a short preview of our findings, our analysis shows that considering both micro-level variables and macro-level contexts is necessary to develop a comprehensive understanding of the emergence of perceptions of technostress and potential channels for mitigation. Analysing novel data from the 2020 wave of the OECD's Risks that Matter (RTM) survey fielded in 24 OECD member countries, this article shows that higher individual income is systematically related to lower levels of perceived technostress. Furthermore, we find that–on average–levels of perceived technostress are lower in countries with a more generous unemployment insurance system, indicating that workers who feel well-protected by the welfare state are less concerned about the stressful impact of technology. Lastly, there is evidence for a cross-level interaction effect between the individual's income position and macro-level contexts. Therefore, a more generous unemployment insurance system further reinforces the negative effect of income on perceived technostress, implying that two resources identified in this article as stress-reducing (income and a generous unemployment insurance system) have mutually reinforcing effects. This multilevel perspective offers a new comparative perspective to the field of technostress, that has mainly considered individual antecedents, but not how cross-national differences might affect technological stress.

## Theory and hypotheses

Technostress is a problem of adaptation that an individual experiences when they are unable to cope with, or get used to, ICTs and is "caused by individuals' attempts and struggles to deal with constantly evolving ICTs and the changing physical, social, and cognitive requirements related to their use" [37, p. 304]. To develop our arguments below, the article continues by explaining the transactional model of stress and coping [24] and its application to the field of technostress [25, 26]. Based on these concepts, we develop arguments on the relationship between income and technostress and whether this relationship is contingent on the welfare state context.

### Appraisal of beneficial or harmful resource conditions

According to the transactional model of stress and coping, individuals engage in a dual cognitive appraisal process [24, 38]. First, they perceive events that might cause stress ("stressors") [24, 39]. In a secondary appraisal, individuals evaluate what resources they have at their disposal to cope with the stressors [24, 40]. If the available resources are sufficient to deal with the stressors, no negative distress occurs; instead, positive eustress develops [39]. However, if resources are insufficient, coping mechanisms are needed to cope with stress.

Lazarus' theory has been used before in literature related to technostress [25, 26], as it emphasizes that stress can result from a combination of demand conditions and individual responses, drawing attention to conditions in which ICT use might be perceived as a negative experience [11, 13]. The individual response to such stressors might depend on a worker's life circumstances and available resources. There could be several possible conditions impacting the technostress experience. First, as job insecurity and technostress are intertwined [41–43], we postulate that income might be a particularly important channel operating on the micro level in buffering technostress. Additionally, we argue that macro-level factors, in particular the institutional set-up of the welfare state, might also matter for technostress perceptions.

### Income and technostress

First, we discuss the relationship between income as an individual resource and technostress perceptions. According to Aljaroodi *et al.* [44], the socioeconomic and technological environments interact and affect each other, e.g., by moderating the process of an ICT user's primary

and secondary appraisal of stress. In this way, higher-income individuals might identify technological stressors in the primary appraisal of stress but also see their financial resources as a valuable tool to cope with these stressors. Therefore, the secondary appraisal of whether there are enough resources available to overcome difficulties with technological stress (e.g., feeling overwhelmed at work and fear of job loss) turns out optimistic.

We argue that individual income helps to buffer potential stress stemming from job insecurity. Job insecurity and technostress are strongly intertwined. On the one hand, job insecurity may lead to increased technostress, as individuals may feel pressure to constantly be available and connected to demonstrate their value to the organisation and avoid being replaced. Job insecurity can increase an individual's stress and anxiety level, making them more susceptible to experiencing technostress [42].

On the other hand, technostress can also contribute to job insecurity. For example, suppose an individual feels overwhelmed by the demands of technology or cannot keep up with rapidly changing technological requirements. In that case, they may feel insecure about performing their job effectively. Employees may worry about being replaced by technology that does their job more efficiently than them. Thus, the fear that ICTs could be taking over their roles may lead to more feelings of job insecurity. For example, Atanasoff and Venable [41] stated that technological demands at the organisational level are associated with an advantage in the labour market, which could indicate that jobs may be at risk due to the rise of ICT. In addition, the perceived pace of technology change positively affects perceived job insecurity due to fear of becoming obsolete or the requirement of learning new skills [43]. We suggest that higher income goes in hand with lower job insecurity, and, thus, with lower perceptions of technostress.

Individuals with higher socioeconomic status have been found to be better equipped with personal resources such as effective coping styles and a reasonable locus of control [21]. Further, an individual's social position can be related to their control over resources [19, 45]. Socioeconomic status has been mostly operationalised as a composite of education and job position in the literature [20]. Education seems to be favourable to the perception of technology: higher education is related to positive perceptions of technology [33]. Additionally, studies that include income as an additional variable for measuring socioeconomic status mostly find that income and education are negatively associated with technostress [22, 23, 46, 47] which might indicate that income is an important confounder. This might support our notion of income being especially relevant for explaining technostress variation. However, several studies also demonstrate that higher job positions and higher education are associated with higher levels of technostress [15–17].

Only a few empirical studies examined whether income is directly associated with general stress or technostress, with inconclusive findings [22, 23, 48–50]. Regarding general stress perceptions (including stress that might not be technology-induced), higher income might be associated with less stress confrontation and an ameliorated psychological health condition [48–50]. Focusing on technostress facets, two studies found that higher monthly income might be associated with lower levels of technostress perception [22, 23]. However, these studies assessing the direct effect of income and technostress were based on online surveys targeting very specific technostress facets, e.g., as in compulsory mobile application use, and suffer some theoretical and methodological issues warranting further research.

We assume that income plays a crucial role in the perception of technostress and argue that the importance of income stems from several channels: Individuals in superior labour market positions are in a better place to keep up with the rapid changes in technology, as they may have more secure labour contracts, find a new job relatively easily, and are, hence, less likely to experience job or wage loss [33, 46, 51]. In addition, people with higher income levels are

more cognisant of the advantages of ICTs, such as the ability to develop social networks and gain wealth [50, 52], potentially indicating more favourable feelings about technology's influence on their jobs [46]. What separates income from other factors, such as education or job positions, which are closely related to income itself, is the financial security it provides. Higher income allows employees to save money and be financially independent for some time, thereby opening up more autonomy in coping with technological changes at work. Hence, it can operate as an individual-level insurance against stressors. Consequently, higher-income individuals may have the necessary resources to cope with technological changes in their working environment and thus have on average lower perceptions of technostress than individuals with lower incomes. In sum, this leads to the following first hypothesis:

*Hypothesis 1*: *Income should be associated with lower levels of perceived technostress.*

## Technostress and the welfare state

In the next step, this article explores the role of macro-level welfare state contexts for the micro-level dynamics of technostress. First, we focus on the direct association between welfare state institutions and individual-level perceptions of technostress, i.e., how the welfare state is related to average stress levels in particular countries. Secondly, we explore to what extent the welfare state mitigates the micro-level association between income and technostress (cross-level interaction effect).

Starting with the first, this study argues that individuals residing in more generous welfare states perceive lower levels of technostress overall. Here, the emphasis is on those dimensions of the welfare state that immediately matter for workers exposed to technostress: the unemployment insurance system. This hypothesis connects to existing work in different ways. For instance, more generous welfare state regimes–such as the Scandinavian countries, but also to some extent the welfare states of Continental Europe–have been found to be associated with better health outcomes and lower health-related inequalities [27–29]. As technostress can also be regarded as a health outcome, we expect a similar dynamic in this case.

More specifically, focusing on labour market policies and their effects on technostress, there is evidence that employment-related policies, such as the generosity of unemployment insurance or the degree of employment protection legislation (EPL) are related to individual perceptions of labour market risk [34–36, 53]. Thus, well-developed labour market policies can be regarded as institutional resources for individuals facing technostress, positively affecting their ability to cope with this stressor. These effects are not necessarily limited to those in precarious employment positions. As shown by Moene and Wallerstein [54], high-income individuals may also support the welfare state as an instrument of social insurance against income losses in the case of unemployment or illness. Hence our first hypothesis regarding the welfare state reads as follows:

*Hypothesis 2a*: *A more generous safety net regarding unemployment insurance policy should be associated with lower levels of technostress across countries.*

Further exploring the links between the welfare state context and the individual level, this article also postulates a novel cross-level interaction between individual income and the generosity of unemployment insurance. In Hypotheses 1 and 2a, we have identified income as individual-level and the welfare state as macro-level resources helping individuals to cope with technostress. When these factors come together, the stress-reducing effects should become mutually reinforcing, suggesting that the (negative) income effect should become even more pronounced in more generous unemployment systems. The theoretical mechanism for our

proposed hypothesis builds again on Moene and Wallerstein's [54] idea that the demand for social insurance can increase with income: high-income earners have relatively more to lose compared to low-income individuals in case of joblessness and hence their demand for insurance increases with income. Relatedly, research also shows that high-income earners are more supportive of social insurance designs in which benefits are provided in relation to previous income [55]. Thus, we argue that high-income individuals residing in welfare states with a more generous unemployment insurance scheme should feel less stressed about technology than high-income individuals in less generous welfare state settings. Put another way, particularly since the amount of unemployment benefits that a person receives is proportional to previous income, a generous unemployment scheme should have a multiplier effect on the technostress-reducing impact of income.

*Hypothesis 2b*: *A more generous safety net regarding unemployment insurance policy should be associated with a larger negative effect of individual income on perceived technostress.*

## Empirical analysis

This study tests the key implications of the theoretical argument using a multilevel modeling strategy. Drawing on original survey questions designed by the authors of this study and included in the most recent wave of the OECD's Risks that Matter (RTM) 2020 survey, the goal is to estimate how the impact of income on technostress varies across different welfare state contexts. The theory predicts that the technostress-reducing effect of income increases with the generosity of the unemployment insurance scheme.

### Measurement

This section explains the measurement of the three main variables of interest–technostress, income, and the welfare state context–and the additional controls in the regression analysis. All the individual-level data used in the analysis are drawn from the aforementioned RTM survey. The RTM survey was fielded in September-October 2020 by the survey contractor Respondi Ltd., which implemented the survey online using non-probability samples recruited via the Internet and over the phone. The RTM survey covers 24,676 individuals aged 18 to 64 years in 24 OECD member countries (Austria, Belgium, Canada, Switzerland, Chile, Germany, Denmark, Spain, Estonia, Finland, France, Greece, Ireland, Italy, South Korea, Lithuania, Mexico, Netherlands, Norway, Poland, Portugal, Slovenia, Turkey, and the United States). The sample is based on quotas for sex, age group, education level, income level, and employment status (in the last quarter of 2019), with the sampling of each category being based on country-specific population data from the OECD to achieve representative quotas for each country in the sample (for additional information, see Box 1.1 in [56]).

**Technostress.** Perceived technostress is measured with a shortened five-item scale based on the original measure by Tarafdar *et al*. [2]. We validated the shortened scale in several steps. First, the Cronbach alpha was sufficient with 0.86, and an exploratory factor analysis clearly indicated a one-factor solution with an eigenvalue of 3.29 and an average item loading of 0.81. Second, a confirmatory factor analysis also resulted in excellent global fit indices for a one-factor solution (CFI = 0.98; TLI = 0.98; RMSEA = 0.03) and an average item loading of 0.76. In particular, we use the following items to measure the five core dimensions of technostress: (1) *techno-overload* ("I am forced by technology to more work than I can handle"), (2) *techno-invasion* ("I feel my personal life is being invaded by technology"), (3) *techno-complexity* ("I often find it too complex for me to understand and use new technologies") (4) *techno-insecurity* ("I feel constant threat to my job security due to new technologies), (5) *techno-*

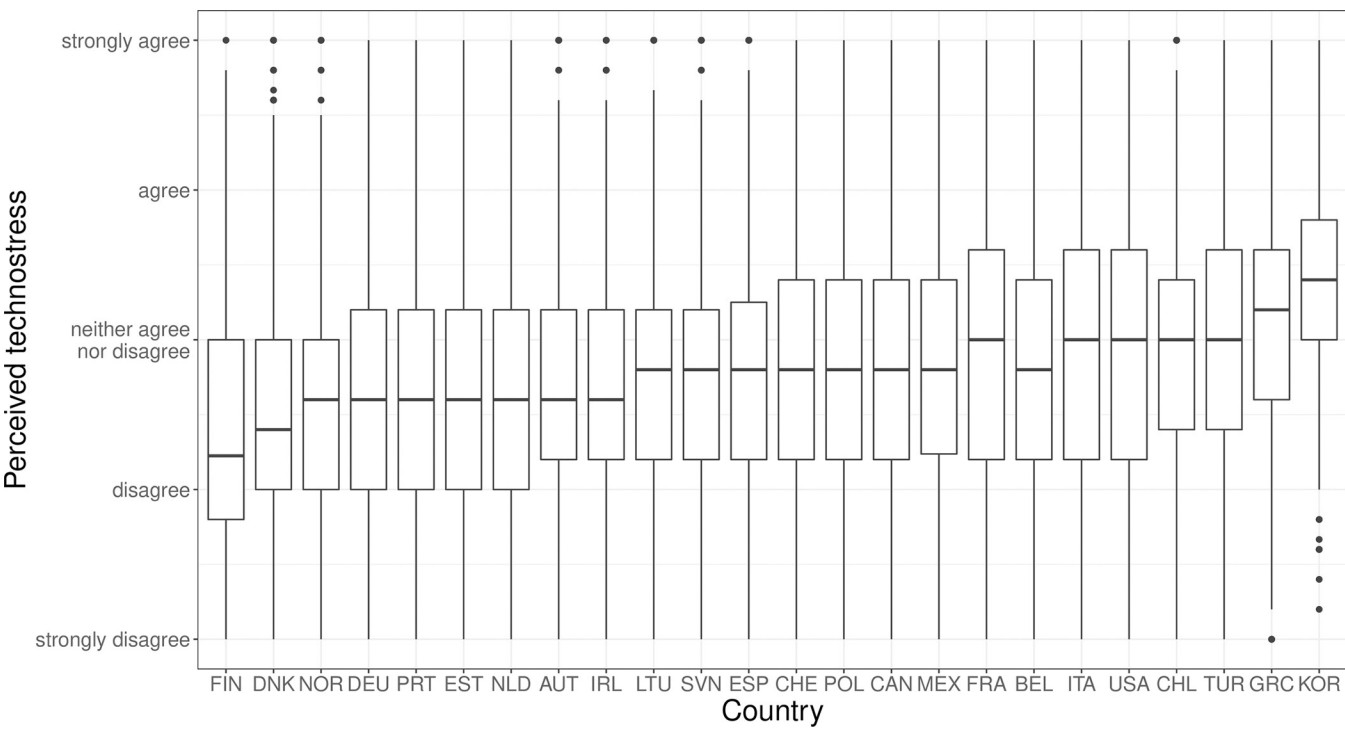

**Fig 1.** Distribution of perceived average technostress across countries.

*uncertainty* ("I perceive that there are always new developments in technologies in my work environment"). Respondents express their agreement to these statements on a Likert scale ranging from "strongly disagree", "disagree", "neither agree nor disagree", "agree" to "strongly agree".

Fig 1 shows the distribution of perceived technostress across countries using country-specific boxplots. The countries are ranked from the lowest to the highest by their average (arithmetic mean) score of perceived technostress. We find the lowest average levels of perceived technostress in some of the Nordic countries (Finland, Denmark, Norway) as well as in some of the core countries of the Eurozone (e.g., Austria, Germany, and the Netherlands). In contrast, perceived technostress appears highest in some of the emerging market economies like South Korea, Turkey, and Chile, in countries of the euro periphery like Italy and Greece, and in the United States. This already indicates some cross-country differences: higher stress levels are to be found in countries with less generous welfare states and/or more liberal labour market regimes, whereas stress levels are lower in the more generous welfare states of Continental and Northern Europe.

**Income.** Income is measured in the RTM survey as the logged disposable annual income equalised for household size. Logging income is a commonly used technique to address the long right tail in income distributions, as the logarithmic transformation makes the distribution more symmetric and thus reduces the impact of extreme values. Purchasing power parities from the OECD are used to standardise incomes across countries to US dollars.

**Welfare state generosity.** To test whether the impact of income on technostress depends on the generosity of the welfare state, we measure welfare state generosity by unemployment benefits, i.e. the existing level of compensation in the case of job loss (this is a common measure for welfare state generosity, see [57]). The data are from the OECD Social and Welfare

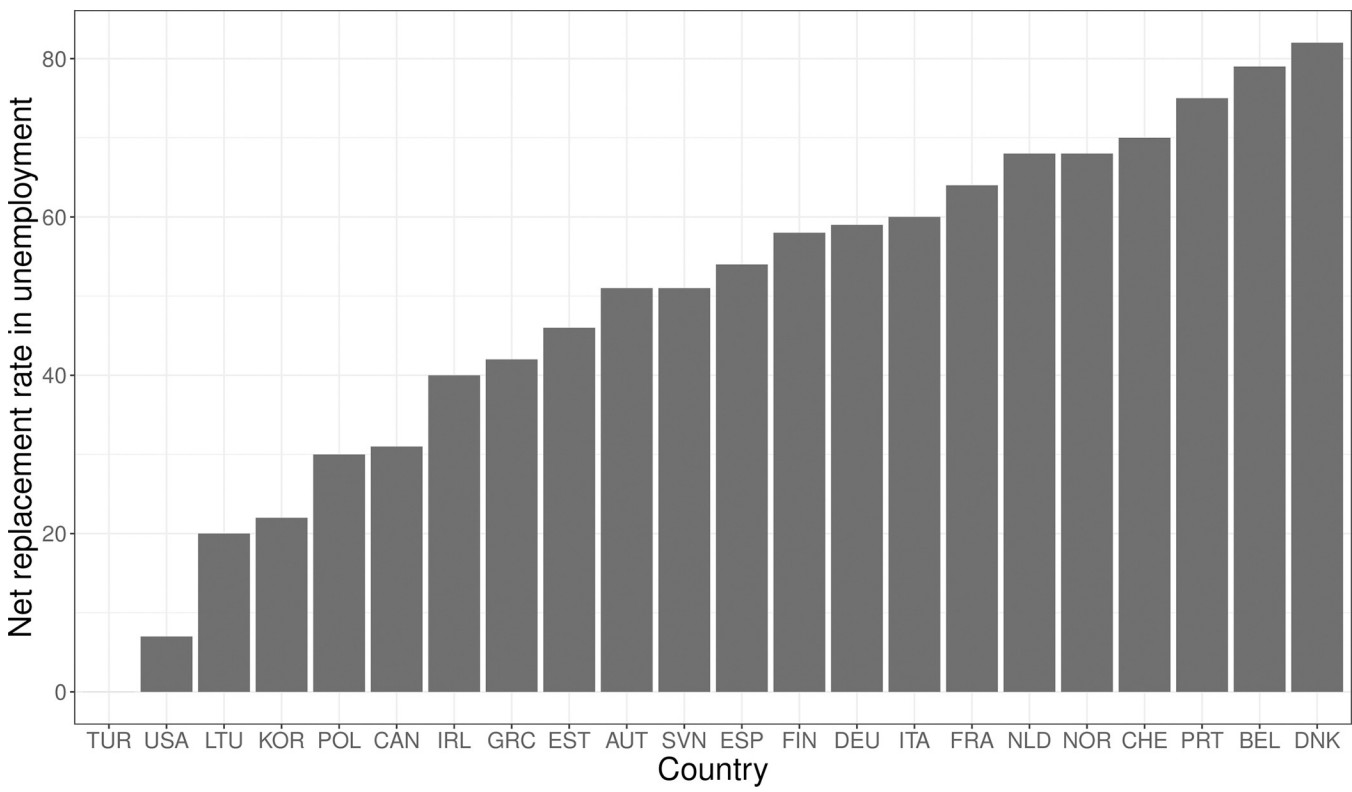

**Fig 2.** Welfare state generosity as measured by unemployment benefits across countries.

Statistics for the years 2019 and 2020 (latest year available) and capture the proportion of previous in-work household income maintained after one year in unemployment (including social assistance benefits). Calculations refer to a single person without children whose last in-work earnings were 67 percent of the average wage. Fig 2 depicts the level of welfare state generosity (ordered from lowest to highest) as measured by unemployment benefits across the countries in the sample. Since data on the level of unemployment benefits are not available for Chile and Mexico, these countries are excluded from the subsequent analysis. The resulting picture broadly confirms that the generosity of unemployment benefits varies along welfare-state regime lines. While the level of generosity is low in emerging economies like Turkey and liberal welfare regimes like the United States, we find high levels of unemployment compensation in established European welfare states like Belgium and Denmark.

**Controls.** This analysis includes both individual-level and country-level control variables in the analysis. On the individual level, we add additional variables from the RTM survey. First, age and a squared term of age are included to account for the possibility that older workers perceive technostress differently than their younger coworkers [26]. A further variable captures the use of information and communication technologies (ICT) at work, as it seems reasonable to expect that technostress is a function of actual technology use [43]. Respondents state whether they use ICT devices (1) never, (2) less than several times a month, (3) several times a month, (4) several times a week, (5) several times a day, or (6) constantly/most of the day. These categories enter the regression analysis as dummy variables, with the first category ("never") serving as the reference category. In addition, the analysis controls for binary indicators for female gender, whether respondents have a child or children, and whether the

respondents have attained tertiary education (previous research has shown that these factors affect technology-related risk perceptions, see [58]).

On the country-level, in addition to the generosity of unemployment benefits, the regression models control for GDP per capita from the OECD National Accounts Statistics in order to account for the possibility that technostress varies across different levels of national development. Moreoever, we include the level of unemployment from the OECD Main Economic Indicators database to control for the current economic situation in a country, assuming that perceptions of stress might be higher in countries with high economic uncertainty. Finally, we use the country-level measure of firm-level technology absorption from the World Economic Forum's Executive Opinion Survey, where business executives were asked to assess to what extent businesses in their respective countries adopt new technology (answers ranged between 1 = "not at all" and 7 = "adopt extensively"). It seems reasonable to expect that the level of technology adoption in a country might correlate with perceptions of technostress.

## Statistical specification and estimation

This section describes how perceived technostress is modeled and how it is shaped by income and the generosity of the welfare state. Since technostress is a latent variable that this study tries to capture through a combination of five indicators from the RTM survey, the analysis uses item response theory (IRT) modeling as a way to define the relationship between observed responses and the underlying latent construct, that is, technostress. More specifically, a one-parameter IRT model is applied, which weights all technostress items equally [59]. Estimating more complex two-parameter IRT models shows that the different technostress items exhibit similar levels of discrimination, a similar pattern of easiness parameters relative to the one-parameter model, as well as a high correlation of country/person parameters between the two models. Moreover, model fit results from approximate leave-one-out cross-validation via Pareto-Smoothed importance sampling do not reveal a clear preference for the two-parameter model. Thus, we rely on the simpler one-parameter model in our analysis. Additionally, the analysis accounts for the fact that individuals are nested in countries by estimating the IRT models in a hierarchical structure. This approach allows us to combine our individual-level data from the RTM survey with information on the country level, in particular the generosity of the welfare state.

The model equation is given by (following the notation for ordered categorical models in McElreath [60], Chapter 12.3):

$$Technostress_{rci} \sim \text{Categorical}(\mathbf{p}_{rci,k})$$

$$\text{logit}(\mathbf{p}_{rci,k}) = \alpha_k - \phi_{rci}$$

$$\phi_{rci} = \beta_0 + \boldsymbol{x}_{rci}\delta + \theta_{r|c} + \vartheta_c + \zeta_i + \varepsilon_{rci},$$

where $Technostress_{rci}$ are the categorical responses of respondent $r$ living in country $c$ to item $i$. The vector $\mathbf{p}_{rci,k} = \{p_{rci,1}, p_{rci,2}, p_{rci,3}, p_{rci,4}\}$ contains the relative probabilities of each response value $k$ (ranging from 1 = "strongly disagree" to 4 = "agree") below the maximum response value of "strongly agree", which by definition has a cumulative probability of 1, for the $r$th respondent from the $c$th country on the $i$th item. The cumulative logit-link function is used to constrain the model predictions to the probability space between 0 and 1. Each response value $k$ is linked to an intercept parameter $\alpha_k$ (i.e., the estimated thresholds between the different ordinal categories) from which the linear model $\phi_{rci}$ is subtracted to ensure that increases in the predictors of this model translate into increases in the average response. In the linear

model itself, $\beta_0$ is the grand mean, $\boldsymbol{x}_{rci}$ is a vector of explanatory variables, in particular individual-specific income, the measure of country-specific welfare state generosity, and the interaction of these two variables, $\theta_{r|c}$ represents the person-specific variance within a given country, $\vartheta_c$ is the country variance parameter, $\zeta_i$ captures the item-specific variance, and $\varepsilon_{rci}$ denotes the error term. Thus, the variance structure reflects that the data vary across individuals nested in countries, and items.

The multilevel ordered logistic IRT models are estimated in a Bayesian framework using the *brms* package in R [61, 62]. Likelihood-based estimation of multilevel models can produce over-optimistic confidence intervals and the problem appears to be particularly severe if the model includes cross-level interactions [63]. In contrast, Monte Carlo evidence suggests that Bayesian estimates of cross-level interactions are more conservative, especially when the number of countries is small [64]. Following the recommendation by Gelman [65] for multilevel models with a small number of groups, priors of the half-*t* family are assigned on the random components. Specifically, we use half-Cauchy priors with $t(4,0,1)$. In addition, all continuous variables are centered and scaled by two times their standard deviation. This makes the standardised coefficients of the continuous variables roughly comparable to the coefficients of the unscaled binary indicators [66].

## Results

We present the empirical findings of our Bayesian multilevel ordered logistic IRT modeling approach in this section. First, we show the results from a model including all our individual-level variables from the RTM survey and all country-level factors, focusing on our variables of interest, i.e., income on the individual level and welfare state generosity on the country level. Next, we report the results from a model that adds the cross-level interaction between these two variables to the previous specification, and calculate and graphically depict quantities of interest in the form of predicted probabilities for this interaction term. Based on our theoretical considerations, we expect that income and the generosity of the welfare state have a negative impact on technostress and that the effect of income grows stronger as welfare state generosity increases.

Fig 3 presents standardised log-odds coefficients (posterior means) and 95% credible intervals based on 6,000 Markov chain Monte Carlo iterations. Regarding our two main variables of interest–income and welfare state generosity, the analysis finds the theoretical expectations corroborated: Both indicators exhibit a statistically significant, negative association with perceived technostress.

To make the interpretation of these findings more intuitive, Fig 4 plots the predicted probability of perceiving technostress conditional on income (Panel A) and welfare state generosity (Panel B), respectively. As expected, the probability of not-perceiving technostress (i.e., to "disagree" or "strongly disagree" with the technostress items) increases strongly with both income (Hypothesis 1) and welfare state generosity (Hypothesis 2a). Looking at the probability to "disagree" with the items measuring technostress, the prediction suggests that at the lowest observed value of income there is roughly a 15 percent probability of disagreement, which increases to more than 50 percent for the highest observed value of income. Regarding welfare state generosity, the same simulation yields an increase from less than 25 percent to roughly 40 percent. Conversely, the probability to "agree" with the technostress items falls from close to 40 percent to less than 10 percent in the case of income and from roughly 25 percent to approximately 15 percent in the case of welfare state generosity.

Turning to the control variables, the analysis finds that the use of ICT devices at work appears to have a nonlinear effect on the perception of technostress (the reference category

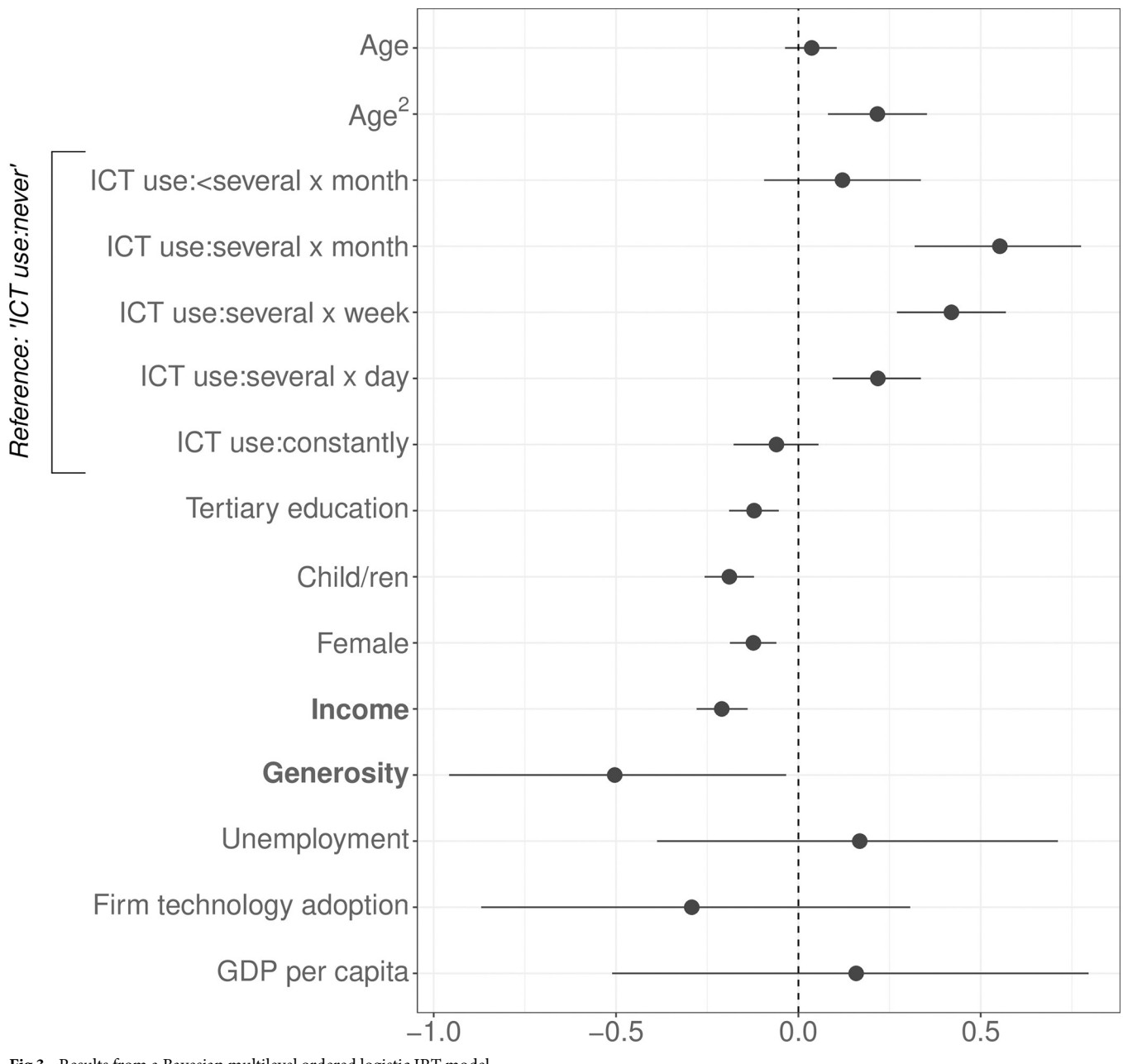

**Fig 3.** Results from a Bayesian multilevel ordered logistic IRT model.

are those individuals who state to "never" use ICT devices at their job). While those workers who use ICT devices several times a month, several times a week, and several times a day perceive significantly higher levels of technostress compared to those who never use these devices, the perception of technostress among those who use ICT devices constantly at work is not statistically significantly different from the reference category. This suggests that the highly tech-savvy workers are also those who feel most comfortable using modern technologies.

In addition, the results show that attaining university-level education, having children, and being female has a statistically significant negative effect on perceiving technostress. At the

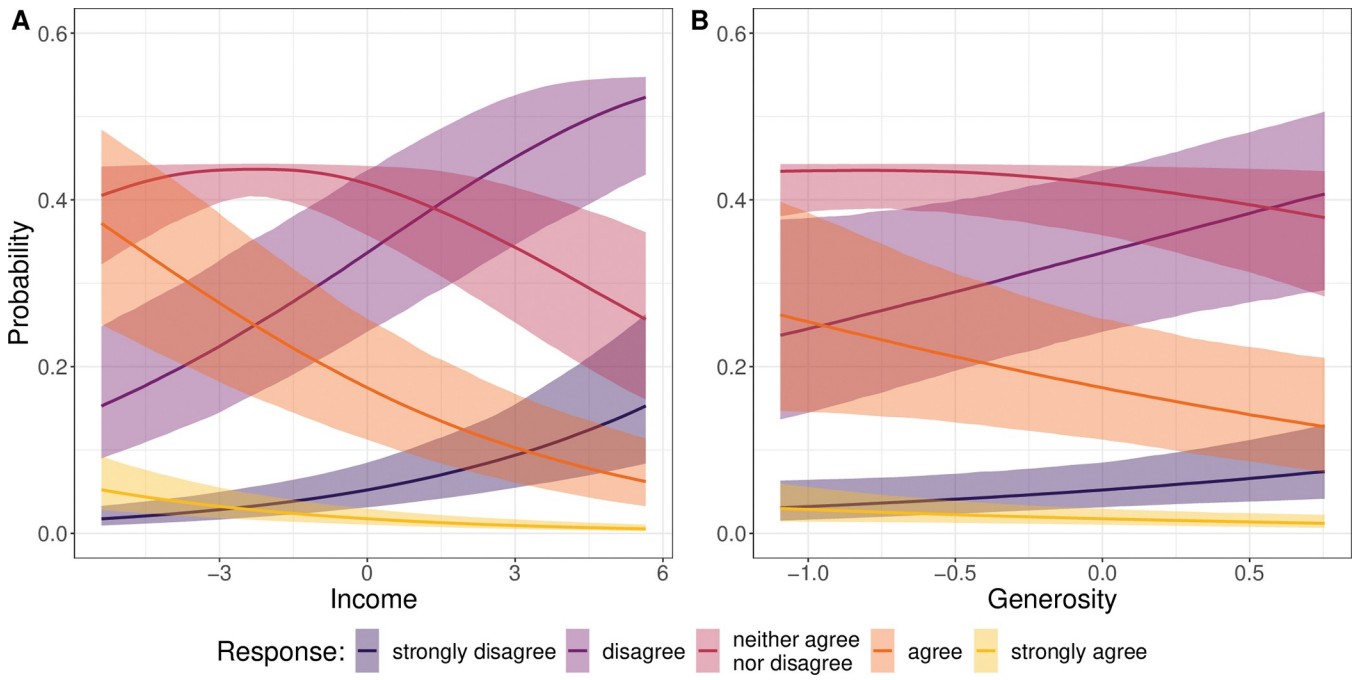

**Fig 4.** Predicted probability of perceiving technostress conditional on income (Panel A) and unemployment generosity (Panel B).

same time, the positive coefficient of the squared term of age suggests that the perception of technostress rises non-linearly with increasing age. None of the country-level control variables reach statistical significance. The theoretical section argues above that the welfare state context–measured by the level of unemployment compensation–affects the micro-level association between income and technostress. More specifically, it is claimed that the negative association between income and technostress increases the more the welfare state insures higher-income earners against potential future income loss in the case of unemployment. Thus, as the existing level of unemployment benefits increases, the effect of income on technostress should become more negative. The analysis estimates a cross-level interaction between income and unemployment benefits to test this argument. The results are depicted in Fig 5.

As theorised, the interactive term between income and unemployment generosity is negative and statistically significantly different from zero. This suggests that in countries with higher levels of generosity the negative association between income and technostress is stronger than in countries with lower levels of unemployment compensation. Again, to gain a more intuitive understanding, Fig 6 presents predicted probabilities for each response value conditional on income, both under low (left panel) and high (right panel) unemployment generosity. Low unemployment generosity is defined as the level of unemployment compensation one standard deviation below the mean and high unemployment generosity as the level of unemployment compensation one standard deviation above the mean. Fig 6 shows that the probability of disagreeing with the technostress items (as income increases) rises markedly more under high generosity than under low generosity of unemployment insurance, in particular in the case of strong disagreement. At the same time, the probability of agreeing to or being indifferent about the presented statements decreases more strongly with income in a high generosity context than under low unemployment generosity. These findings underscore that the negative effect of income on technostress is amplified in a generous welfare state context, as stated by Hypothesis 2b.

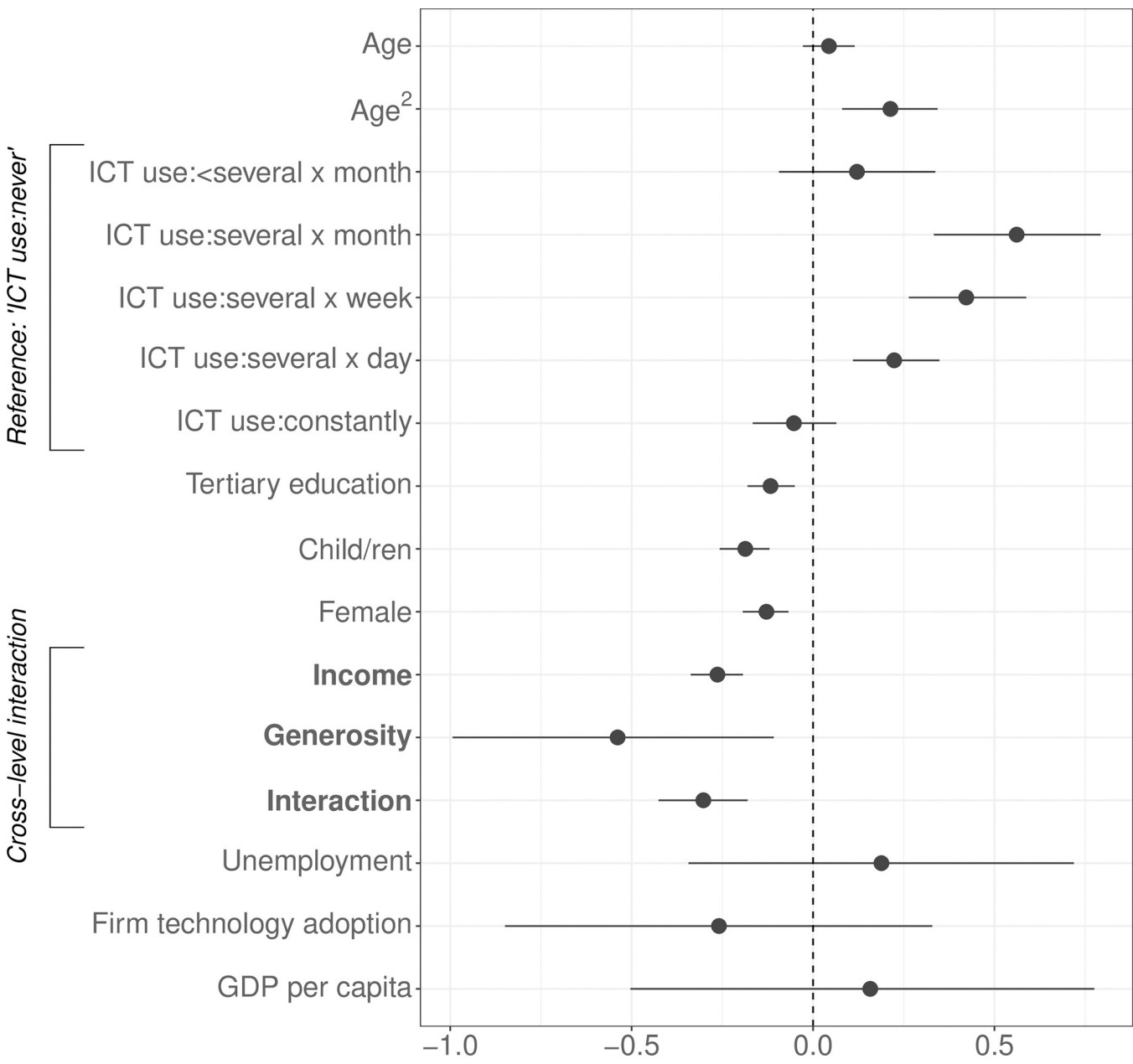

**Fig 5.** Results from a Bayesian multilevel ordered logistic IRT model with cross-level interaction between income and unemployment generosity.

## Discussion

This study investigated whether there is a relationship between income and technostress at work and how contingent this relationship is on the welfare state context. We analysed novel and original data from the latest wave of OECD's RTM survey by applying Bayesian multilevel ordered logistic IRT models that take into account the latent character of perceived technostress and the hierarchical nature of the dataset. Our results showed that both income and unemployment generosity were negatively related to perceived technostress. This corroborates the argument that both factors serve as a resource helping individuals cope with perceived workplace technostress. Moreover, the article provided evidence suggesting that the negative

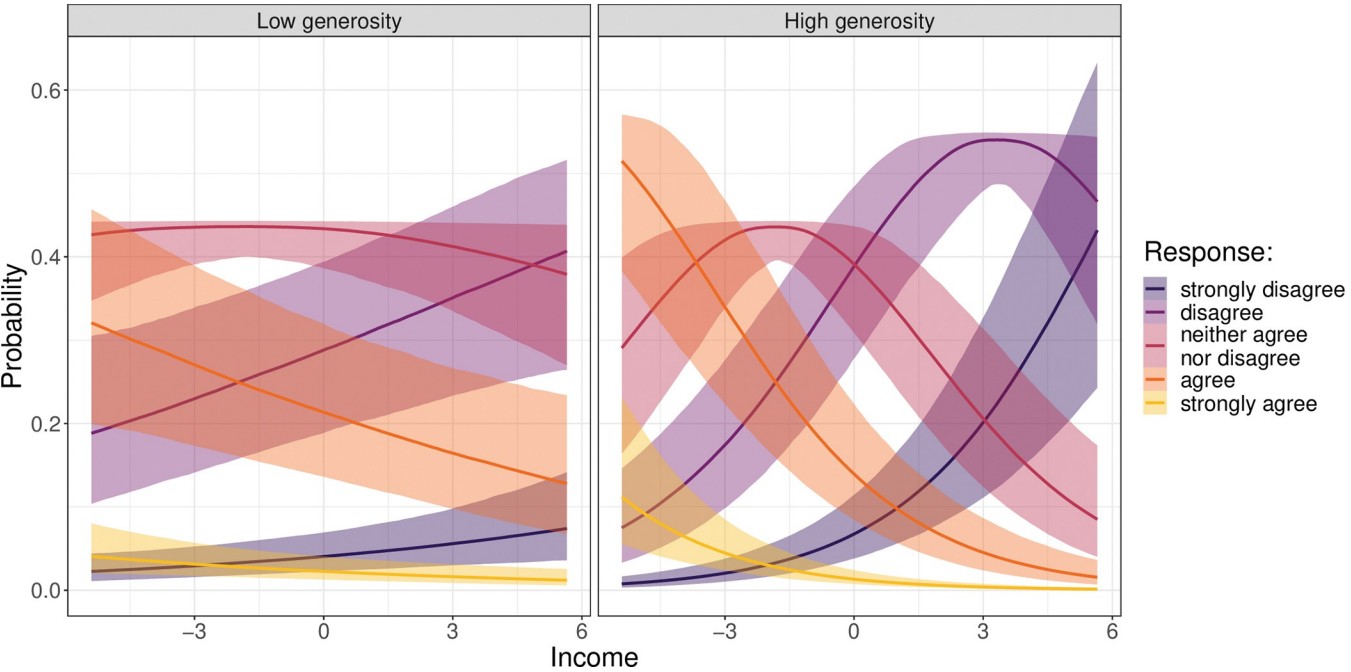

**Fig 6.** Predicted probability of perceiving technostress conditional on income under low and high unemployment generosity.

effect of income on technostress increases as unemployment generosity rises, supporting the proposed hypothesis that unemployment benefits are an important insurance tool against income loss in the case of job loss for higher-income earners and thus amplify the technostress-reducing effect of income.

## Theoretical and practical implications

Based on these findings, this article extends the implications of the relationship between income, technostress and welfare state support in multiple ways. A first insight here is that income could be a channel for mitigating the perception of technostress. Its financial security and, thus, lower job insecurity, might provide individuals with more resources (second appraisal) to cope with the stressors identified (first appraisal). Our results that higher earners experience less technostress are counterintuitive, considering previous research stating that highly educated workers with higher job-positions experience more technostress [15–17]. At the same time, it is in line with previous research stating that highly educated individuals experience less technostress [30], e.g. due to effective coping mechanisms [21]. Further, our results highlight previous research saying that an individual's social position can be related to their control over resources [19, 43], as income might be one of the resource factors buffering technostress. In this way, our study might provide evidence that enough available income is a resource to deal with technological stressors, buffering negative distress [39], and that it is worth it to study income's relationship to technostress separately from other socioeconomic status factors. Further, organisations should be aware that there are several ways to buffer technostress, e.g, by facilitating ICT literacy, keeping users informed about the rationale for introducing new ICTs, and offering a supportive organisational culture to prevent perceived technostress in the workplace [4, 5, 67].

Second, our results indicate that the welfare state context does matter. We highlight that technostress is an intriguing factor in the investigation of inequalities in health determinants

between different welfare systems. In doing so, we build on earlier research on technology-related anxiety [33] and risk perceptions associated with technological change [30–32], now with a clear focus on technostress as a health indicator. Our study indicates that more generous welfare states, in particular labour market policies, can dampen subjective perceptions of technostress, providing an additional resource that helps individuals to cope with stress. In doing so, we want to put the importance of various structural elements on the theoretical technostress agenda. Our multilevel viewpoint provides an innovative comparative perspective on technostress, an area that has mostly focused on individual and organisational antecedents (see [68]) rather than how cross-national differences may impact technological stress.

In our study, the welfare state context also has ambivalent implications regarding inequality. Our analysis shows that particularly high-income individuals benefit relatively more from the buffering effects of welfare state institutions. Hence, in terms of perceived stress, the welfare state aggravates income-related inequalities in perceived stress. In other words, technostress could increase the current job polarisation between low-skilled and low-income versus high-skilled and high-income workers, fostering job inequity. At the same time, according to our findings, welfare states could help to buffer this polarisation by offering stable unemployment compensation. Consequently, social policies should be seen as essential to offer resources for people to cope with rising levels of technology in the workplace. Labour market policies should be designed in ways that are particularly targeted at low-income recipients, for instance, by lowering eligibility thresholds and relaxing means-testing procedures.

## Limitations and future research

This study has several limitations. First of all, technostress has been defined in various ways in previous research and also in this study we cannot clearly distinguish between technostress at work and technostress in the personal sphere. However, as the boundaries between work and life are blurring in today's working world, such a strict distinction might not necessarily be realistic. Second, our measures capture subjective perceptions of technostress, which might be different from objective measures of technostress, i.e., actually observed health outcomes. The latter are inherently difficult to measure, especially in cross-country studies, such as ours, regarding how to separate tech-related stressors from other sources of stress related to employment or personal circumstances. Nevertheless, it would be important to better understand how objective health outcomes (such as cortisol-level or heart-rate variability) are related to subjectively perceived technostress.

## Conclusion

The article uses original OECD survey data to explore the impact of information and communication technology overload on workers, leading to a condition known as technostress, and emphasizes the importance of coping resources. It introduces a novel approach by linking the welfare state context to how income levels affect technostress, revealing that higher earners perceive less technostress. The study suggests that welfare state benefits, particularly unemployment benefits, may mitigate technostress by providing a safety net.

## Author Contributions

**Conceptualization:** Ann S. Lauterbach, Florian Kunze, Marius R. Busemeyer.

**Data curation:** Tobias Tober, Marius R. Busemeyer.

**Formal analysis:** Tobias Tober.

**Funding acquisition:** Florian Kunze, Marius R. Busemeyer.

**Methodology:** Tobias Tober.

**Project administration:** Ann S. Lauterbach.

**Supervision:** Florian Kunze, Marius R. Busemeyer.

**Visualization:** Tobias Tober.

**Writing – original draft:** Ann S. Lauterbach, Tobias Tober, Marius R. Busemeyer.

**Writing – review & editing:** Ann S. Lauterbach, Florian Kunze.

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
