## [Decision Letter · Decision Letter 0]

11 May 2023

PONE-D-22-33306Can welfare states buffer technostress? Income and technostress in the context of various OECD countries.PLOS ONE

Dear Dr. Lauterbach,

Thank you for submitting your manuscript to PLOS ONE. After careful consideration, we feel that it has merit but does not fully meet PLOS ONE’s publication criteria as it currently stands. Therefore, we invite you to submit a revised version of the manuscript that addresses the points raised during the review process.

Please note that we have only been able to secure a single reviewer to assess your manuscript. We are issuing a decision on your manuscript at this point to prevent further delays in the evaluation of your manuscript. Please be aware that the editor who handles your revised manuscript might find it necessary to invite additional reviewers to assess this work once the revised manuscript is submitted. However, we will aim to proceed on the basis of this single review if possible. While overall positive towards publication, the reviewer has provided some comments regarding opportunities for clarification and elaboration. Please ensure you address each of the reviewer's comments when revising your manuscript.

We look forward to receiving your revised manuscript.

Kind regards,

Hugh Cowley

Staff Editor

PLOS ONE

2. Peer review at PLOS ONE is not double-blinded (https://journals.plos.org/plosone/s/editorial-and-peer-review-process). For this reason, authors should include in the revised manuscript all the information removed for blind review.

“The authors acknowledge the funding by the Deutsche Forschungsgemeinschaft (DFG—German Research Foundation) under Germany’s Excellence Strategy (Grant Number EXC2035/1-390681379).”

4. Please amend your authorship list in your manuscript file to include author names.

Reviewers' comments:

Reviewer's Responses to Questions

**Comments to the Author**

1. Is the manuscript technically sound, and do the data support the conclusions?

Reviewer #1: Yes

2. Has the statistical analysis been performed appropriately and rigorously? 

Reviewer #1: Yes

3. Have the authors made all data underlying the findings in their manuscript fully available?

Reviewer #1: Yes

4. Is the manuscript presented in an intelligible fashion and written in standard English?

Reviewer #1: Yes

5. Review Comments to the Author

Reviewer #1: This is a well written and executed study on income, technostress, and welfare state protection. It will make a nice contribution to the literature. I have a few minor points the authors may want to consider in their revision.

1. Until I got to the methods section where the measure of technostress is described, I was wondering where job insecurity fits into all of this. I think it would be helpful to add a paragraph or two that describes the relationship between job insecurity and technostress.

2. It would be helpful if you included the actual scale that the individual components of the technostress latent variable were measure on.

3. It may be worth noting why income is logged.

4. I found it a bit odd that in the Results section the authors began with a discussion of the findings of the control variables and then the discussion of the findings relevant to the research questions followed. I would consider switching this.

6. PLOS authors have the option to publish the peer review history of their article (what does this mean?). If published, this will include your full peer review and any attached files.

Reviewer #1: No

---

## [Author Response · Author response to Decision Letter 0]

24 Jun 2023

1. Is the manuscript technically sound, and do the data support the conclusions?

Comment #1: Yes

Our answer: Thank you for your positive feedback.

2. Has the statistical analysis been performed appropriately and rigorously?

Comment #2: Yes

Our answer: Thank you for the encouragement of our method.

3. Have the authors made all data underlying the findings in their manuscript fully available?

Comment #3: Yes

Our answer: Thank you for acknowledging our data access.

4. Is the manuscript presented in an intelligible fashion and written in standard English?

Comment #4: Yes

Our answer: Again, we thank you for the affirmation that we have created a sound contribution.

5. Review Comments to the Author

Comment #5.0: This is a well written and executed study on income, technostress, and welfare state protection. It will make a nice contribution to the literature. I have a few minor points the authors may want to consider in their revision.

Our answer: Thank you very much for your detailed feedback. We will address all of your specific concerns thoroughly below.

Comment #5.1: Until I got to the methods section where the measure of technostress is described, I was wondering where job insecurity fits into all of this. I think it would be helpful to add a paragraph or two that describes the relationship between job insecurity and technostress.

Our answer: Thank you for this remark on the theoretical framework. We agree that our argumentation for selecting job insecurity as a mechanism was too shortly explained in the theory part of the prior version. Therefore we added more details on this in the hypotheses development (pp. 7-8) section.

Comment #5.2: It would be helpful if you included the actual scale that the individual components of the technostress latent variable were measure on.

Our answer: We added the original scale to Fig 1. 

Comment #5.3: It may be worth noting why income is logged.

Our answer: Thank you for this comment. Based on your recommendation, we added a footnote (new Footnote 2) explaining the rationale for logging income.

Comment #5.4: I found it a bit odd that in the Results section the authors began with a discussion of the findings of the control variables and then the discussion of the findings relevant to the research questions followed. I would consider switching this.

Our answer: Thank you for this remark. We changed the order of the results section to improve the reader’s understanding. 

Finally, we want to thank you again for the diligent review of our paper and the concrete ideas on how to improve it. We hope that you generally appreciate our revision.

---

## [Decision Letter · Decision Letter 1]

23 Aug 2023

PONE-D-22-33306R1Can welfare states buffer technostress? Income and technostress in the context of various OECD countries.PLOS ONE

Dear Dr. Lauterbach,

Thank you for submitting your manuscript to PLOS ONE. After careful consideration, we feel that it has merit but does not fully meet PLOS ONE’s publication criteria as it currently stands. Therefore, we invite you to submit a revised version of the manuscript that addresses the points raised during the review process.

We look forward to receiving your revised manuscript.

Kind regards,

Adeel Luqman

Academic Editor

PLOS ONE

Reviewers' comments:

Reviewer's Responses to Questions

**Comments to the Author**

1. If the authors have adequately addressed your comments raised in a previous round of review and you feel that this manuscript is now acceptable for publication, you may indicate that here to bypass the “Comments to the Author” section, enter your conflict of interest statement in the “Confidential to Editor” section, and submit your "Accept" recommendation.

Reviewer #1: All comments have been addressed

Reviewer #2: (No Response)

2. Is the manuscript technically sound, and do the data support the conclusions?

Reviewer #1: Yes

Reviewer #2: No

3. Has the statistical analysis been performed appropriately and rigorously? 

Reviewer #1: Yes

Reviewer #2: No

4. Have the authors made all data underlying the findings in their manuscript fully available?

Reviewer #1: Yes

Reviewer #2: Yes

5. Is the manuscript presented in an intelligible fashion and written in standard English?

Reviewer #1: Yes

Reviewer #2: No

6. Review Comments to the Author

Reviewer #1: The authors have done a nice job addressing my comments. I look forward to seeing it in publication.

Reviewer #2: The study reveals that the authors aimed to explore the influence of the welfare state context on the correlation between income and technostress. Nevertheless, the study could benefit from a restructuring. The introduction section would benefit from the inclusion of research motivation and the identification of research gaps. The theory section lacks in-depth discussion. Notably, the stress appraisal theory does not align with some of the proposed relationships in the study, such as that between income and technostress. Furthermore, the discussion in the hypothesis development section diverges from the proposed hypotheses. It would be advisable for the authors to reconsider the research model in accordance with a more appropriate theory. Enhancements are also needed in the research methodology section, particularly regarding insights into the data collection process, respondent profiles, and the criteria employed for respondent selection.

Moreover, the study indicates the utilization of both primary and secondary data. The author could present robust arguments, accompanied by citations, demonstrating the suitability of utilizing data gathered from both primary and secondary sources for constructing a research model. However, there appears to be a lack of clarity regarding the source of measurement for the income variable. Additionally, it is advisable for the control variables to be substantiated by referencing prior literature. The statistical techniques employed should be properly justified in relation to the collected data. Finally, there is room for enhancement in the sections covering results, discussion, and conclusions.

7. PLOS authors have the option to publish the peer review history of their article (what does this mean?). If published, this will include your full peer review and any attached files.

Reviewer #1: No

Reviewer #2: No

---

## [Author Response · Author response to Decision Letter 1]

4 Oct 2023

We would like to express our sincere thanks to Prof. Luqman and our two reviewers for the valuable comments and suggestions we received after the first round of review. We appreciate that you felt confident enough in our research to allow us to revise and resubmit our work again. Your comments and feedback were insightful and developmental. Thank you for taking the time.

We have worked through the reviews and made substantial revisions to the manuscript to respond to your comments and suggestions. We want to summarize the main adjustments here:

(1) First, we followed the feedback regarding the restructuring and readability of our study by a) stating a more detailed research motivation (p. 4), b) providing more context and literature to embed the hypothesis development in our theoretical stress framework (e.g., p. 7-9), and c) delving deeper into the discussion and comparing our results to previous research (p. 22-23). 

(2) We also addressed all the methodological concerns raised by reviewer 2 and provide now, for example, more details on our data (p. 13) and the reasoning behind the choice of certain control variables (p. 16). Further, we explained why the hierarchical Item Response Theory (IRT) model allows us to combine our individual-level data with information on the country level (p. 17).

Finally, we made several other minor changes to the manuscript, which are explained in detail in the response to reviewers document, when answering your specific requests point by point.

---

## [Decision Letter · Decision Letter 2]

23 Oct 2023

PONE-D-22-33306R2Can welfare states buffer technostress? Income and technostress in the context of various OECD countries.PLOS ONE

Dear Dr. Lauterbach,

Thank you for submitting your manuscript to PLOS ONE. After careful consideration, we feel that it has merit but does not fully meet PLOS ONE’s publication criteria as it currently stands. Therefore, we invite you to submit a revised version of the manuscript that addresses the points raised during the review process.

We look forward to receiving your revised manuscript.

Kind regards,

Adeel Luqman

Academic Editor

PLOS ONE

Journal Requirements:

Reviewers' comments:

Reviewer's Responses to Questions

**Comments to the Author**

1. If the authors have adequately addressed your comments raised in a previous round of review and you feel that this manuscript is now acceptable for publication, you may indicate that here to bypass the “Comments to the Author” section, enter your conflict of interest statement in the “Confidential to Editor” section, and submit your "Accept" recommendation.

Reviewer #2: (No Response)

2. Is the manuscript technically sound, and do the data support the conclusions?

Reviewer #2: Partly

3. Has the statistical analysis been performed appropriately and rigorously? 

Reviewer #2: Yes

4. Have the authors made all data underlying the findings in their manuscript fully available?

Reviewer #2: Yes

5. Is the manuscript presented in an intelligible fashion and written in standard English?

Reviewer #2: Yes

6. Review Comments to the Author

Reviewer #2: Thank you for re-inviting me to review the manuscript entitled "Can welfare states buffer technostress, Income and technostress in the context of various OECD countries". The author made a good effort. However, I still want to the author to consider the following:

1. Lines 8 and 9 in the "income and technostress" section appear to be confusing. To clarify, please provide an explanation supported by scholarly citations or a logical argument regarding the role of income in the first and second appraisal of stress.

2. Please fortify hypothesis 2b with appropriate scholarly citations.

3. Provide a more detailed explanation of quota sampling, including the population proportion allocated to each quota. In case uncontrolled quota sampling is used, be sure to indicate this in the study limitations.

4. Clearly state which indicators were utilized to measure "Welfare State Generosity" and explain how these indicators were managed in the construction of the "Welfare State Generosity" scale.

5. Address how missing data for Chile and Mexico were handled during the analysis.

6. Consider separating the "implications" section from the "discussion and conclusion" section.

7. Evaluate if the study carries theoretical implications and, if so, include them in the paper following the discussion section.

7. PLOS authors have the option to publish the peer review history of their article (what does this mean?). If published, this will include your full peer review and any attached files.

Reviewer #2: No

---

## [Author Response · Author response to Decision Letter 2]

10 Nov 2023

We would like to express our sincere thanks to Prof. Luqman and our reviewers for the valuable comments and suggestions we received after the second round of review. We appreciate that you felt confident enough in our research to allow us to revise and resubmit our work again. Thank you for taking the time.

We have worked through the reviews and made several changes to the manuscript, which are explained in detail in the following sections, when answering your specific requests point by point.

Editorial Comments

Comment #1: Please review your reference list to ensure that it is complete and correct. If you have cited papers that have been retracted, please include the rationale for doing so in the manuscript text, or remove these references and replace them with relevant current references. Any changes to the reference list should be mentioned in the rebuttal letter that accompanies your revised manuscript. If you need to cite a retracted article, indicate the article’s retracted status in the References list and also include a citation and full reference for the retraction notice.

Our answer: Thank you for pointing this out. We meticulously reviewed our reference list and confirmed that no article was retracted using the website http://retractiondatabase.org/RetractionSearch.aspx

Regarding new references, we included the following references to support our revised version: 

Fetscher, Verena (2023). Explaining support for redistribution: social insurance systems and fairness. Political Science Research and Methods, 11, 588-604.

Nisafani, A. S., Kiely, G., & Mahony, C. (2020). Workers’ technostress: A review of its causes, strains, inhibitors, and impacts. Journal of Decision Systems, 29(1), 243-258.

OECD (2021). Main Findings from the 2020 Risks that Matter Survey, OECD Publishing, Paris.

Scruggs, Lyle A. and Gabriela Ramalho Tafoya (2022). Fifty years of welfare state generosity. Social Policy & Administration, 56(5), 791-807.

Comments Reviewer 2

Comment #1: Thank you for re-inviting me to review the manuscript entitled "Can welfare states buffer technostress, Income and technostress in the context of various OECD countries". The author made a good effort. However, I still want to the author to consider the following. 

Our answer: Thank you very much for your detailed feedback. We will address all of your specific concerns thoroughly below.

Comment #2: Lines 8 and 9 in the "income and technostress" section appear to be confusing. To clarify, please provide an explanation supported by scholarly citations or a logical argument regarding the role of income in the first and second appraisal of stress.

Our answer: Thank you for pointing this out. We carefully revised and slightly restructured the relevant section in order to bring more clarity to our argumentation. We now start the argument with the following statement (p. 7f):

“ … higher-income individuals might identify technological stressors in the primary appraisal of stress but also see their financial resources as a valuable tool to cope with these stressors. Therefore, the secondary appraisal of whether there are enough resources available to overcome difficulties with technological stress (e.g., feeling overwhelmed at work and fear of job loss) turns out optimistic.” 

This theoretical presentation is followed by the argumentation about job insecurity and technostress as before. Finally, there are two sections on the empirical evidence to date (p. 9f): first on socioeconomic status and technostress in general, then more specifically on income and stress or technostress. In this way, the current evidence is emphasized more clearly with the appropriate references.

Comment #3: Please fortify hypothesis 2b with appropriate scholarly citations.

Our answer: Thank you for this remark on hypothesis 2b. Since the proposed cross-level interactive relationship is -- to our best knowledge -- novel, there is no authoritative literature that we could draw on. In the revised version, we, therefore highlight the novelty of the argument and calarify that the hypothesis is introduced by us (p. 12). Moreover, we added an recently published reference, which shows empirically that high-income earners are more supportive of insurance designs if the benefits are provided in relation to previous income, as this finding reinforces the underlying mechanism of our proposed hypothesis.

Comment #4: Provide a more detailed explanation of quota sampling, including the population proportion allocated to each quota. In case uncontrolled quota sampling is used, be sure to indicate this in the study limitations.

Our answer: To clarify these aspects, we added more information on the quota sampling, which is based on representative population data for each country. We also included a reference to the main OECD report that provides a description of the sampling procedure (see p.13).

Comment #5: Clearly state which indicators were utilized to measure "Welfare State Generosity" and explain how these indicators were managed in the construction of the "Welfare State Generosity" scale.

Our answer: Thank you for this note. In the revised version of the paper, we state more clearly that welfare state generosity is measured by a country's unemployment benefits (net replacement rates of previous income). We also added a reference (Scruggs & Gabriela, 2022) that underscores that this is a common measure of welfare state generosity in the literature (p. 15).

Comment #6: Address how missing data for Chile and Mexico were handled during the analysis.

Our answer: Thank you for pointing this out. Since data for Chile and Mexico are not available in our dataset, these two countries are not included in the analysis. The revised version of the paper states this more explicitly (p. 15).

Comment #7: Consider separating the "implications" section from the "discussion and conclusion" section.

Our answer: We agree that the final section of our paper could still be better structured. Therefore, we divided the section into two main sections “discussion” and “conclusion”, whereby “discussion” was divided into subsections including the summary of the main findings, theoretical and practical implications, and limitions and future research. In addition, we reordered the sections according to these new headings (see p. 23-25).

Comment #8: Evaluate if the study carries theoretical implications and, if so, include them in the paper following the discussion section.

Our answer: Thank you for this remark. We revised the implications part of the discussion section and clarified where we see our theoretical contribution. We emphasize that our study is the first multilevel perspective on the technostress research field, which has so far focused strongly on the individual and organizational level (p. 24, paragraph 1). Further, we stress that technostress has so far received too little attention in comparative welfare state research - although such stress is increasingly becoming the norm due to advancing technological developments in the work environment (p. 24, paragraph 2).

Finally, thank you again for the diligent and patient review of our paper and the concrete ideas for improving it. We really feel confident about our current version and hope that it brought us closer to a publication in PLOS ONE.

---

## [Decision Letter · Decision Letter 3]

20 Nov 2023

Can welfare states buffer technostress? Income and technostress in the context of various OECD countries.

PONE-D-22-33306R3

Dear Dr. Ann Sophie Lauterbach,

We’re pleased to inform you that your manuscript has been judged scientifically suitable for publication and will be formally accepted for publication once it meets all outstanding technical requirements.

Kind regards,

Adeel Luqman

Academic Editor

PLOS ONE

Additional Editor Comments (optional):

Reviewers' comments:

Reviewer's Responses to Questions

**Comments to the Author**

1. If the authors have adequately addressed your comments raised in a previous round of review and you feel that this manuscript is now acceptable for publication, you may indicate that here to bypass the “Comments to the Author” section, enter your conflict of interest statement in the “Confidential to Editor” section, and submit your "Accept" recommendation.

Reviewer #2: All comments have been addressed

2. Is the manuscript technically sound, and do the data support the conclusions?

Reviewer #2: Yes

3. Has the statistical analysis been performed appropriately and rigorously? 

Reviewer #2: Yes

4. Have the authors made all data underlying the findings in their manuscript fully available?

Reviewer #2: Yes

5. Is the manuscript presented in an intelligible fashion and written in standard English?

Reviewer #2: Yes

6. Review Comments to the Author

Reviewer #2: Thank you for the revision. However, kindly ensure that proofreading is conducted before submitting the final version of the manuscript for publication.

7. PLOS authors have the option to publish the peer review history of their article (what does this mean?). If published, this will include your full peer review and any attached files.

Reviewer #2: No

---

## [Editor Report · Acceptance letter]

27 Nov 2023

PONE-D-22-33306R3 

Can welfare states buffer technostress? Income and technostress in the context of various OECD countries. 

Dear Dr. Lauterbach:

I'm pleased to inform you that your manuscript has been deemed suitable for publication in PLOS ONE. Congratulations! Your manuscript is now with our production department. 

Kind regards, 

on behalf of

Dr. Adeel Luqman 

Academic Editor

PLOS ONE